# Using Metaheuristics (SA-MCSDN) Optimized for Multi-Controller Placement in Software-Defined Networking

**Neamah S. Radam *** , **Sufyan T. Faraj Al-Janabi** and **Khalid Sh. Jasim**

College of Computer Science and Information Technology, University of Anbar, Ramadi 31001, Iraq
* Correspondence: nea19c1010@uoanbar.edu.iq

**Abstract:** The multi-controller placement problem (MCPP) represents one of the most challenging issues in software-defined networks (SDNs). High-efficiency and scalable optimized solutions can be achieved for a given position in such networks, thereby enhancing various aspects of programmability, configuration, and construction. In this paper, we propose a model called simulated annealing for multi-controllers in SDN (SA-MCSDN) to solve the problem of placing multiple controllers in appropriate locations by considering estimated distances and distribution times among the controllers, as well as between controllers and switches (C2S). We simulated the proposed mathematical model using Network Simulator NS3 in the Linux Ubuntu environment to extract the performance results. We then compared the results of this single-solution algorithm with those obtained by our previously proposed multi-solution harmony search particle swarm optimization (HS-PSO) algorithm. The results reveal interesting aspects of each type of solution. We found that the proposed model works better than previously proposed models, according to some of the metrics upon which the network relies to achieve optimal performance. The metrics considered in this work are propagation delay, round-trip time (RTT), matrix of time session (TS), average delay, reliability, throughput, cost, and fitness value. The simulation results presented herein reveal that the proposed model achieves high reliability and satisfactory throughput with a short access time standard, addressing the issues of scalability and flexibility and achieving high performance to support network efficiency.

**Keywords:** MCPP; SA-MCSDN; HS-PSO; virtual machine; NS3

## 1. Introduction

SDNs are in a constant state of evolution as new mechanisms emerge; therefore, it is critical to solve scalability issues by taking full advantage of the programmability of controllers, without sacrificing performance and management capabilities.

In this work, an optimized simulated annealing (SA)-based algorithm (SA-MCSDN) is proposed to effectively deploy multiple controllers in order to reduce connection latency and propagation and improve throughput and reliability. First, the network is generated; then, an optimal controller is selected in terms of its features using the FA algorithm. Finally, multiple controllers are placed based on the selected controller using the proposed algorithm, which increases the convergence rate, reducing deployment latency and connection latency.

Simulated annealing (SA) is a commonly used algorithm based on the principle of the heuristic algorithm, determining the optimal solution by comparing possible solutions. Thus, SA can be used to solve the multi-controller placement problem (MCPP), which involves placing controllers in appropriate positions in order to achieve the shortest distances and the least communication time between multi-controller interfaces, as well as between controllers and switches.

The SA method involves the following aspects:

- Annealing: Refers to heating a solid to a sufficiently high temperature so that its molecules are arranged randomly. Then, the temperature is gradually reduced, so once cooled, the molecules of the solid are arranged in a lower-energy stable state.
- Heating processing: Whereby the thermal movement of particles is enhanced, eliminating any non-uniform state that may exist in the system.
- Thermal processing: In closed systems that exchange heat with the environment without a change in temperature, the state of the system is altered without constraint as free energy diminishes. When free energy reaches a minimum value, the system achieves a state of harmony.
- Cooling processing: The thermal movement and arrangement of particles are gradually debilitated, and the vitality of the system is gradually reduced, resulting in a low-energy crystal structure. SA is based on the principles of statistical mechanics.

The remainder of this paper is organized as follows. In Section 2, we review related works, and a statement of the research problem is presented in Section 3. In Section 4, we describe the proposed SA-MCSDN method in terms of network construction, optimal controller selection, and multi-controller placement. The experimental results of the proposed SA-MCSDN approach are explained in Section 5. Finally, in Section 6, we present our conclusion on the proposed SA-MCSDN approach and present some suggestions for future work.

## 2. Related Work

The authors of [1] proposed an optimization solution for the controller placement problem (CPP) called SA failure foresight capacitated controller placement problem (SA-FFCCPP). The authors of [2] proposed and implemented a greedy heuristic annealing simulation algorithm to solve the CPP. The first task is to decide where to place the controllers within a resource-limited network. The proposed algorithm determined that digital controllers require that all network elements be covered in an optimal manner. The primary criterion employed by the algorithm was to reduce the distance between all nodes and selected controllers. The authors of [3] presented a solution based on the simulation of an annealing algorithm, whereby feedback resulted in significant improvements, representing a specific implementation of resource planning with a focus on a resource mapping model.

The authors of [4] used machine learning in various application scenarios for network control and optimization, for example, network traffic analysis, traffic forecasting, abnormal analysis, network simulation, and fault diagnosis. Path optimization algorithms include particle swarm algorithms, genetic algorithms, and SA algorithms. The authors of [5] decomposed the intractable offline problem into smaller cases, which were solved in an efficient manner online using an algorithm based on SA. Furthermore, they analyzed the need for frequent adjustment of the control plane and compared multiple control plane design options according to a new elasticity scale. The authors of [6] proposed and experimentally tested a method based on SA and quantitative annealing (QA), which surpassed other methods in terms of performance and synchronization cost. In addition, relative to integer linear programming (ILP), their proposed algorithm is significantly more scalable, making it applicable to large-scale networks.

The authors of [7] proposed a multi-controller hierarchical deployment strategy for a space–air–ground integrated network for the sixth generation (6G SAGIN) of an SDN-based system. A multiple-propagation strategy was employed to determine the delay model of the network, the loading model of the SDN controller, and the loss value to be optimized based on the SA to search for the optimal solution space. The proposed strategy takes into account dynamic changes in network topology, as well as SDN controller imbalances, to improve network performance. The authors of [8] proposed a solution to improve throughput, the probability of link failure, and transparency on the southbound interface (SBI) for control-level synchronous transmission in a wireless controller mode (WCPP), whereby the southbound interface (SBI) depends on an unlicensed 4G LTE long-term

evolution network. Two indicative solutions were considered: one based on SA and the other based on radiography.

The authors of [9] evaluated the performance of Cuckoo-PC in comparison with SA and quantitative softening (QA) methods. Experiments showed that Cuckoo-PC surpasses both (SA) and QA in terms of network performance. In [10], a QoS service was developed in an avionics system operating in a heterogeneous wireless network environment by applying two algorithms—analytical hierarchy process (AHP-SA) and SA for synchronous weights and optimizing network selection (SA-SWNO)—to dynamically optimize the weighting factors of objective functions. The authors of [11] focused on C2C and S2C delays resulting from link failure and unreliability associated with the CPP. They used two algorithms—spectral clustering and ant colony (ACO)—to generate a versatile assignment plot for computing resources. They determined that the resource allocation algorithm achieved better performance than the ant colony algorithm, according to the criteria described above, and showed that SA diverged from the ideal neighborhood arrangement of the optimal local solution with a certain probability.

The authors of [12] proposed SA based on enhanced mass peak density (DBSAA). Experiments were conducted on a real-world network topology to verify the effectiveness of the proposed algorithm for common placement problems, achieving nearly optimum performance in a shorter runtime. In [13], a heuristic update algorithm (HRIA) based on (SA) and a greedy algorithm was proposed to search for an approximate optimal solution. The authors of [14] proposed simulated double annealing; their results showed that the double-annealing algorithm outperformed the latest hybrid annealing and aggregation algorithm in terms of solution accuracy, with a small tradeoff in terms of runtime. The authors of [15] proposed an SDN controller deployment scheme based on a simulated solid genetic hybrid algorithm. This algorithm combines SA and the genetic algorithm. SA is used to quickly search for an improved solution and quickly find the optimal path. The authors of [16] introduced a strategy to address the CPP that protects against latency, potential link failure, and transparency in the case of an SBI wireless interface. They modelled the problem of locating wireless controllers in an SDN. To this end, the authors presented an indicative solution based on SA and genetic algorithms (GAs) that provides a fast and efficient solution.

The authors of [17] suggested a hybrid of PSO and SA called HPSOSA for a controller flex mode to reduce the mean latency, as controller flex mode deals with the issue of node failure or link failure. The authors of [18] proposed a multi-objective hybrid harmony algorithm (MOHS-SA) to identify optimally Pareto-distributed solutions, for example, to reduce the total operating costs of facilities, optimize vehicle mileage, and reduce the cost of $CO_2$ emissions. The authors of [19] proposed a hybrid algorithm (HS-SA) that considers dynamic values of harmony memory (HMCR) and pitch modulation rate (PMR) using local optimization techniques for hybridization and probability based on SA. The authors of [20] also presented a hybridization simplification algorithm that combines HS and SA, known as HS-SA, for accurate and precise breast malignancy detection.

The authors of [21] proposed a control system based on the integration of SDN and IoT in Smart City environments, based on application requirements. This control system detects when an emergency occurs and dynamically modifies the routes of normal and emergency urban traffic in order to reduce the time required for emergency resources to arrive at the emergency area, achieving an improvement ratio of 33%—from 26 ms to 17 ms. The proposed control system works through the application of priority and path-sharing avoidance via an alert treatment algorithm. An average time difference of 50% was achieved between the alternative path and the congested path. The authors of [22] proposed a dynamic QoS algorithm in an SDN to select the optimum path that ensures video QoS in order to optimize the quality of experience (QoE). The results demonstrate that the proposed method achieved improved viewing quality for a smart community environment and increased the overall network throughput.

The authors of [23,24] proposed a hybrid metaheuristic HSA-PSO algorithm to effectively deploy multiple controllers according to an MCSDN approach in order to reduce communication and propagation latency and improve throughput and reliability.

Table 1 provides a summary of the weaknesses, limitations, and objectives of the surveyed literature, which can inform the research findings of the proposed algorithm (SA-MCSDN) in terms of various performance measures.

**Table 1.** Comparison of the proposed SA-MCSDN algorithm with previous works.

| Study Ref./No | Drawbacks and Weaknesses of Literature Review | Objective(s) | SA-MCSDN Approach |
|---|---|---|---|
| [1] | Focus only on latency, reducing worst-case latency between S2C, and reducing the execution time for large networks; failures and limitations on control capabilities are expected, and this model produces a near-perfect solution. The futuristic view of the researcher is that this work can be expanded by making the position of the controllers and the task of switching energetic. Moreover, controller latency can be included within the objective function and considered as a load-balancing aspect. | SA-FFCCPP-based optimized solution for CPP in SDN. | The MCPP was resolved for distribution simulation, and execution time between S2C and C2C was solved for all metrics. See Table 2. |
| [2] | Handles a limited number of resources within the network, and digital controllers require that all network elements within the network be covered in an optimal manner. The primary criterion was to reduce the distance between all nodes and selected controllers by a greedy SA algorithm. | Implement a greedy SA algorithm to solve the CPP; the goal was to reduce the distance between all selected nodes and controllers. | The primary criterion was to reduce the distance between all nodes and selected controllers and to reduce the execution time by verifying all metrics. |
| [3] | Focused on the resource mapping model. | Feedback-sensitive resource mapping based on an SA algorithm in an SDN. | SA-MCSDN was used to solve the MCPP. |
| [4] | Used machine learning and focused only on network traffic analysis, traffic prediction, and fault diagnosis. | SA algorithm used for path optimization for network control. | SA-MCSDN was used to solve the MCPP. |
| [5] | Focused only on the flexibility dynamic control plane (DCPP) to reduce the total cost without using metrics for measure scalability. | Flexible design of DCPP in an SDN by an online SA algorithm. | SA-MCSDN was used to solve the MCPP. |
| [6] | SA and QA performance is significantly more scalable with higher synchronization cost. | Multi-objective placement evolution of SDN controllers to improve cost and network performance in WSN. | The SA-MCSDN method was used to solve the MCPP in terms of metrics, performance, and cost. |
| [7] | One approximate comparison method was used (n-k-means vs. n times k-means) running a time clustering algorithm, which increased performance by 17.7%, whereas the controller optimized load dynamic strategy (COLDS) increased performance by approximately 7.71%. | Multi-controller deployment of 6G SAGIN in an SDN by an SA algorithm. | The running time of the algorithm (SA-MCSDN) increased by 3.33%. |
| [8] | Fourth-generation long-term evolution (4G LTE-Unlicensed) was implemented with LTE-U-CPP-RS ray-shooting heuristic; the simulation results revealed better and more accurate results than the LTE-U-CPP-SA heuristic. | The use of LTE-U-CPP-SA and LTE-U-CPP-RS algorithms to solve WCPP based on a 4G LTE network with SBI. | SA-MCSDN was used to solve the MCPP. |
| [9] | Experiments showed that Cuckoo-PC surpasses both SA and QA in terms of a range of performance indicators. Cuckoo-PC accomplishes less than 1% deviation in a discernibly shorter time than SA. | Cuckoo placement of controllers (Cuckoo-PC algorithm) in an SDN, employing SA and QA to optimize the network performance in WSNs. | SA-MCSDN was used to solve the MCPP. |
| [10] | By learning with AHP-SA and SA-SWNO, the execution time can be reduced to optimize the weights of factors. | SA-based multilink choice calculation algorithm in SDN-enabled avionic networks. | Without learning, SA-MCSDN can decrease the execution time and encourage optimization of the components. |
| [11] | Focused on delay, C2C, and S2C from where link failure originated, as well as reliability for CPP. Used two algorithms (spectral clustering and ant colony) to generate an adaptive allocation scheme for computing resources. | Dynamic placement of MCSDN and allocation of computational resources based on a heuristic ant colony algorithm. | According to the results of the SA-MCSDN optimized algorithm, it achieved the best performance solving the problem of delay and reliability. |

**Table 1.** *Cont.*

| Study Ref./No | Drawbacks and Weaknesses of Literature Review | Objective(s) | SA-MCSDN Approach |
|---|---|---|---|
| [12] | Used DBSAA to achieve nearly optimum performance in a shorter runtime for the joint placement problem of gateways and controllers. | Joint placement of gateways and controllers in an SDN-enabled space–ground integration network. | The SA-MCSDN algorithm processes the appropriate positioning of the controllers of C2C and S2C position and handles the delay time and reliability, i.e., to extend the network efficiency and performance. |
| [13] | Used HRIA based on SA and a greedy algorithm to search for the approximate optimal solution and to alleviate the high time cost of the rule-update process. | To achieve fast ternary content, addressable memory (TCAM) update with BatchUp processing optimization in an SDN for two stages. | The SA-MCSDN algorithm was used to search for the optimal performance with a shorter runtime time of the update process and a shorter distance between S2C and C2C. |
| [14] | Used a combination of GA and SA with a clustering hybrid algorithm. Used double SA to quickly search for a better solution and quickly find the optimal path to achieve improved flexibility, scalability, reliability, and latency with a minimal tradeoff in terms of runtime. | SA of joint controller and gateway placement in 5G-satellite SDN networks. | The SA-MCSDN single optimized algorithm was used to search for the optimal performance with less runtime. |
| [15] | The SA-GA hybrid algorithm solves the CPP by locating wireless controllers in an SDN. It can reduce the cost of cost controller deployment by guaranteeing the delay between S2Cs reduces deployment time. | Optimization of SDN controller deployment based on an SA-GA hybrid algorithm. | The SA-MCSDN single algorithm achieved a short delay time guarantee for communication between S2C and C2C with deployment based on FA. |
| [16] | Used SA and GA to locate wireless controllers and solve the CPP in an SDN using metrics of latency, potential link failure, and transparency in the case of an SBI wireless interface S2C. | CPP for a wireless software-defined network (WSDN). | The SA-MCSDN algorithm was optimized to solve the MCPP between S2C and C2C. |
| [17] | Used the HPSOSA algorithm to solve CPP-SDN flex mode and minimize the average latency to address the issue of node failure or link failure. | HPSOSA hybrid approach to solve CP-SDN. | The SA-MCSDN single algorithm was optimized to solve the MCPP between S2C and C2C. |
| [18] | MOHS-SA algorithm was used to find the optimally Pareto-distributed solutions, for example, to reduce the total cost of operating facilities, improve vehicle mileage, and reduce the cost of $CO_2$ emissions. | Multi-objective hybrid algorithm (MOHS-SA) to solve a location-inventory-routing issue in the supply chain arrangement plan of inverted coordinates with $CO_2$ emissions. | The SA-MCSDN algorithm was optimized to solve the MCPP between S2C and C2C. |
| [19] | Used a hybrid (HS-SA) algorithm that considers dynamic values of HMCR and PMR with local optimization techniques for hybridization and probability based on SA. | Hybrid algorithm(HS-SA) to solve a location-inventory-routing issue in a supply chain network plan (SCN) with deformity and non-defect objects. | The SA-MCSDN single algorithm was optimized for probability Pr ($\Delta$F). |
| [20] | Used the HS-SA algorithm for precise and accurate breast cancer detection in an intelligent healthcare system for optimized diagnosis. | An intelligent healthcare framework to optimize breast cancer screening using an HS-SA algorithm. | The SA-MCSDN algorithm was optimized to solve the MCPP between S2C and C2C. |
| [21] | Focused on energy-consuming traffic management in emergency situations. | SDN-based control system for efficient traffic management for emergency situations in smart cities. | SA-MCSDN was used to solve the MCPP. |
| [22] | Used a flow-based routing strategy for video service routing with a focus on addressing two constraints: packet loss and bandwidth. | A QoS-based routing algorithm in an SDN for video surveillance. | SA-MCSDN was used to solve the MCPP. |

**Table 2.** Average results of the proposed SA-MCSDN algorithm.

| Performance Metrics | SA-FFCCPP | HSA-PSO | SA-MCSDN |
|---|---|---|---|
| Propagation delay | 31.3 | 15.6 | 4.80 ms |
| Average RTT | 10.95 | 7.5 | 6.05 ms |
| Matrix of time session | 246.5 | 203.5 | 113.13 |
| Average delay | 95.3 | 78.8 | 42.57 ms |
| Reliability | 79% | 87% | 99% |
| Throughput (response/s) | 165.71 | 197.14 | 395.57 Kbps |
| Cost | 37.5 | 27.4 | 31.27 |
| Fitness value | 18.2 | 22.5 | 19.16 |

## 3. Problem Statement

The MCPP is one of the crucial issues in SDNs. The placement and location of the multi-controllers affect the performance efficiency of the network. There are many challenges encountered during controller placement, such as the propagation delay between S2Cs, the tolerance of controller faults, and meeting switch requirements. Therefore, it is important to ensure reduced network propagation time, fewer errors in link failure or failure of the node itself, and reduced cost to deploy devices or nodes in the network, as well as improved reliability and throughput. Many models have been proposed for simulation and design, for example, a framework and a mathematical model that contain many algorithms to check the exactness of the proposed mathematical model and verify the correct performance of the proposed algorithm. Through simulations, a comparison can be made with the model for the MCPP [25].

Traffic-aware models, system-aware models, and rule-placement models consist of objective mathematical formulation model solutions [26]—for example, the mathematical model for planning the deployment of an SDN using a new ILP mathematical model; given several input parameters, the model has two distinct capabilities [27]. There are four basic requirements for sequential implementation, as illustrated in Figure 1.

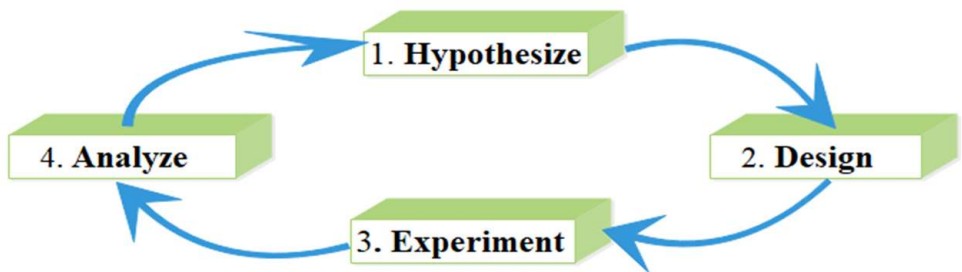

**Figure 1.** The iterative process of the scientific method [28].

## 4. The Proposed Work

The proposed method is adaptable, reliable, and improves the general performance of the network. The methodology used in this research is based on the following three processes: (1) building and configuring the network settings based on space topology and the virtual devices used on the network, as shown in the proposed model in Figure 2, which is a simplified diagram of the proposed MCSDN architecture; (2) a mechanism of distribution and selection of controllers using the firefly algorithm (FA); and (3) the use of our proposed algorithm (SA-MCSDN) to choose the most suitable locations for the controllers. The current research is a continuation of previous work under the title Multi-Controllers Placement Optimization in SDN by the hybrid HSA-PSO algorithm, in which we used an algorithm to solve the same problem of MCSDN to select suitable locations for controllers. Table 2 shows the results from our previous work, as well as a comparison between our previous work and this research.

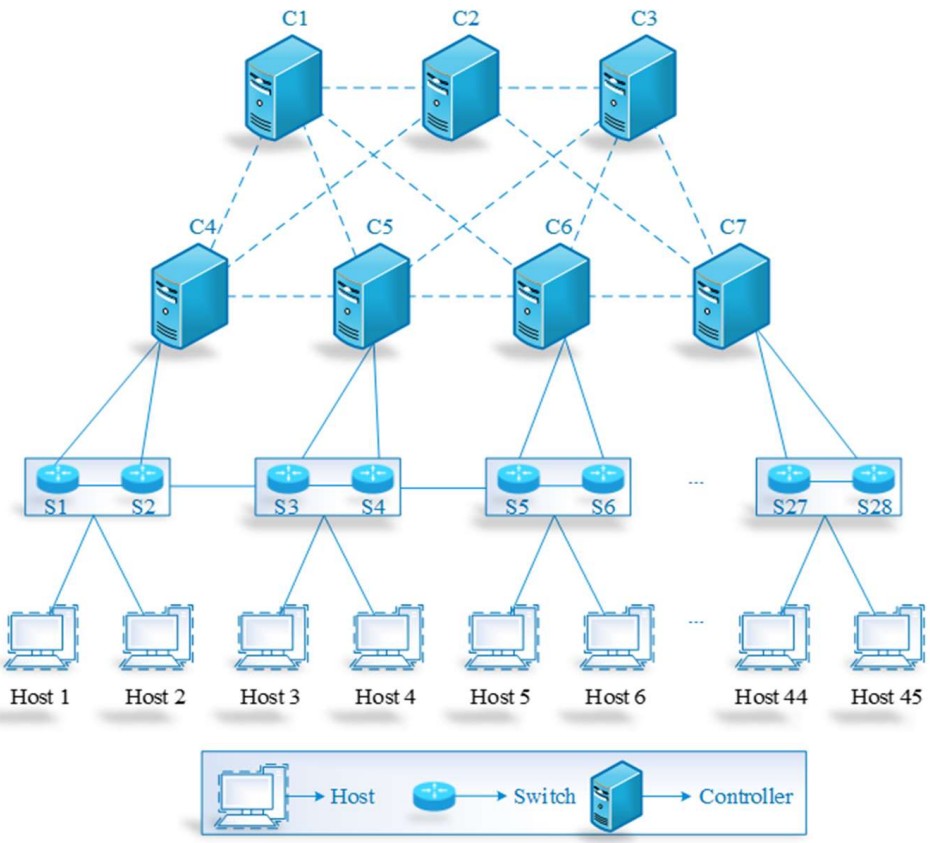

**Figure 2.** Multi-controller SDN simulation environment.

Figure 2 represents the system model, for which the mathematical equations in this section are formulated as follows: seven (*n*) controllers were used in the network topology, with twenty-eight (*n*) switches and forty-five (*n*) hosts.

*4.1. Network Construction*

The topology is built based on the undirected graph structure [29], as expressed in the following:

$$G = (V, E, U)$$

(1)

Let us assume $C = (c_1, c_2, \ldots, c_n)$, $S = (s_1, s_2, \ldots, s_n)$, $V = C \cup S$, $n = V$, $k = U$, and $P_c = P_{c1}, P_{c2}, \ldots, P_{cm}$

$$m = \frac{n!}{k!(n-k)!}$$

(2)

Table 3 contains a description of the graph theory, topology, and other mathematical symbols [30,31].

**Table 3.** Description of graph theory, topology, and other mathematical symbols.

| Symbol | Description |
|---|---|
| $G = (V, E, U)$ | Physical network topology graph |
| $V$ | The node set in the network topology set of $n$ switches: data-plane nodes |
| $E$ | Set of physical links between switches (edges) |
| $U$ | Set of $k$ controllers, where $k = U$ |
| $C$ | Set of controllers, where $C \subset V$ |
| $(i, j)$ | The link between node $i$ and node $j$ |
| $d (S, C)$ | The distance between switch $S \in V$ and controller $C \in V$ |
| $n$ | Total number of switches or nodes in the network elements, where n = V |
| $k$ | Total number of controller clusters to be installed in the network, where $k \leq$ n |
| $Pc$ | Set of all possible placements for $k$ controllers (probability): the probability of network component failure including nodes and links |
| $xi;k$ | *Indicates whether switch i is associated with controller k (= 1) or not (= 0)* |
| $X$ | Geographic locations of nodes |
| $Sec$ | Seconds |
| $M$ | Meters |
| $ms$ | Milliseconds |
| $j$ | Latitude of a node |
| $F_{Cn}$ | Firefly controller of $n$ |
| $F_C$ | Controller features |
| $D$ | Distance |
| $D_C$ | Distance between controllers |
| $\aleph_0$ | Maximum light intensity for firefly |
| $FT(D)$ | Fault tolerance of distance |
| $\aleph(D)$ | Controller absorption coefficient based on FA |
| $\gamma$ | Absorption coefficient |
| $\aleph$ | Optimal controller |
| $m$ | Number of all possible placements of $(n)$ elements |

### 4.2. Optimal Controller Selection

The process of FA-based controller selection is illustrated by the pseudocode presented below. First, the controller features within the network are initialized, as formulated in Algorithm 1 [24].

---

**Algorithm 1** Pseudocode FA for Optimal Controller Selection

---

Initialize: $F_{Cn} = \{F_{C1}, F_{C2}, F_{C3}, F_{C4}, F_{C5}, F_{C6}, F_{C7}, s .., F_{Cn}\}$ ;
Formulate objective function using: $FT_{F_C} = f(F_C)$;
Formulate controller absorption coefficient using: $\aleph(D) = \aleph_0 e^{-\gamma D^2}$;
Initialize absorption coefficient using:
$$\aleph(FT_{F_C}) = \{\aleph(FT_{F_{C1}}), \aleph(FT_{F_{C2}}), \ldots, \aleph(FT_{F_{C7}})\} ;$$
**While** all $F_C$ **do**
Compute controller feature distance using: $FT(D) = \frac{F_C}{D_C}$ ;
**For** $F_{C1} = 1$ to n (all n controllers)
**For** $F_{C2} = 1$ to n (all n controllers)
**If** ($D_{F_{C2}} > D_{F_{C1}}$), select $F_{C2}$ over $F_{C1}$
**End if**
Update the $\aleph(D)$
**End for** $F_{C2}$
**End for** $F_{C1}$
Rank the controller and find the current best controller
**End while**

---

### 4.3. Multi-Controller Placement Using Simulated Annealing Algorithm

In the SDN environment, the optimal positioning of controllers reduces latency during communication between the controllers and switches. Therefore, we adopted a simulated annealing algorithm. The SA algorithm is a global optimization algorithm inspired by the

thermodynamic process. The thermodynamic process reduces the energy of a material and changes its state through a mechanism of controlled heating and cooling in order to change the physical properties of the material according to its free thermodynamic energy. The heating and cooling process affects the temperature and thermodynamic free energy of the substance. The idea of slow cooling involves the temperature being gradually decreased from an initial positive value to zero in order to rank solutions. Specifically, SA metaheuristics involve approximation of a global optimization in a large search space for an optimization problem. The state of the physical system and the function to be minimized are analogous to the internal energy of the system. The goal is to transform the system from a random initial state to a minimal energy state. SA can be used for very difficult computational optimization problems, for which precise algorithms fail. Although SA usually achieves only an approximate solution to the global minimum, it may suffice for many practical problems.

This phenomenon results in an optimization problem for engineering domains. The adoption of SA yields increased accuracy and provides the optimal solution with a slower convergence speed. In our work, a highly accurate and robust optimized method is used for optimal placement of multiple controllers in the SDN environment. SA can solve non-linear global problems, as it depends on a stochastic model. The objective function ($F(x)$) is provided in maximized and minimized forms below [1,4,10,19].

In the maximized form, the objective function can be formulated as

$$F(\overrightarrow{x_{OPT}}) = \max_{\overrightarrow{x_i} \in Y} F(\overrightarrow{x}) \tag{3}$$

In Equation (3), the objective function is represented by the optimal vector, which solves the problem of placing controllers at appropriate locations in the network in the case of the maximum function represented by $F(\overrightarrow{x_{OPT}})$ of $x_{i,j}$.

In the minimized form, the objective function can be formulated as

$$F(\overrightarrow{x_{OPT}}) = \min_{\overrightarrow{x_i} \in Y} F(\overrightarrow{x}) \tag{4}$$

In Equation (4), the used objective function is represented by the optimal vector, which solves the problem of placing controllers in appropriate positions in the case of the minimum function represented by $F(\overrightarrow{x_{OPT}})$ of $x_{i,j}$.

From the above equations, $\overrightarrow{x}_i$ is the variable obtained by SA (i.e., $\overrightarrow{x}_i \in \varphi$). Some of the steps of SA are provided below.

Phase 1: An initial temperature is set to generate a random solution ($x_{i,j}$), which can be formulated as

$$\forall_i = x'^{\text{ maxi}}_T + \forall_i * \left( x'^{\text{ maxi}}_T - x^{\text{ mini}}_{T0} \right) \tag{5}$$

where $\forall_i \in [1, M]$, in which M defines the temperature, and $x'^{\text{ maxi}}_T$ and $x^{\text{ mini}}_{T0}$ are the ending and starting temperatures, respectively. The above equation computes the fitness value of $x_{i,j}$.

Equation (5) is used to calculate the fitness value between upper and lower cases for $f(x)$ of the optimal vector ($x_{i,j}$).

Phase 2: Based on the present point, random feasible neighbor points are generated. Random point generation continues until a feasible neighbor point is estimated ($x'_{i,j}$). Then, based on the above equation, the fitness function is computed to calculate the difference between the two fitness functions as

$$\Delta F = \left( F\left(x'_{i,j}\right)_{\text{New}} - F\left(x_{i,j}\right)_{\text{Current}} \right) \tag{6}$$

Equation (6) is used to calculate the density function ($\Delta F$) difference between the two fitness functions.

Phase 3: In this step, a new optimal point is selected. If $\Delta F < 0$, $\left(x'_{i,j}\right)_{New}$ and $\left(x_{i,j}\right)_{Current}$ are the new optimal points, and $F\left(x'_{i,j}\right)_{New} = F\left(x_{i,j}\right)_{Current}$ is used for the next process. Otherwise, the probability density function $(Pr(\Delta F))$ is computed, which can be formulated as

$$Pr(\Delta F) = exp^{\left(\frac{-\Delta F}{T_{L(itr)}}\right)} \tag{7}$$

Equation (7) is used to calculate the probability of the density function $Pr(\Delta F)$, which is a standard equation for all parameters.

Then, a random number $(\Theta)$ is generated to obtain the optimal position. If $\Theta$ is less than $Pr(\Delta F)$, then $\left(x'_{i,j}\right)_{New}$ is the optimal point of $\left(x_{i,j}\right)_{Current}$, and the next phase proceeds. Otherwise, the process returns to phase 2. The formulation is expressed as

$$Pr = \begin{cases} \left(x'_{i,j}\right)_{New}, & \text{if } \Theta\epsilon[0,1] < Pr(\Delta F) \\ \Delta F = F\left(x'_{i,j}\right)_{New} - F\left(x_{i,j}\right)_{Current}, & Else \end{cases}, \tag{8}$$

The probability of Equation (8) consists of two parts: (1) the position of the controllers in terms of distance and time and (2) the density function $(\Delta F)$.

Phase 4: **If** $L < K$, **then** put $L = L + 1$ and jump to phase 2. **Else if**, if $L > K$, stop the searching process, and else go to phase 5.

Phase 5: Put $L = L + 1$, $L = 1$; put

$$T_L = \Theta T_{L-1} \tag{9}$$

Equation (9) is used to calculate the temperature with the Boltzmann constant; the temperature reduction factor (cooling rate) increases with each step.

Phase 6: **If** $S' >= $ Worst Solution (S) **then** accept the best worst solution (S); **Else if** calculate

$$T0 = \frac{Worst - Best}{log(\beta)} \tag{10}$$

$$T_{itr+1} = \alpha \times T_{itr}, \tag{11}$$

Equation (10) is used to calculate the initial temperature depending on the best and worst solutions. Equation (11) is calculated after completing the optimal solution, which represents the position of the first controller; here, a new iteration is added to proceed to the second step.

itr = itr + 1; and jump to phase 2.

Through the above six steps, the following final formula is obtained:

$$\text{Maximize } \sum_{i=1}^{n} \alpha_{i,j} - \frac{1}{2} \sum_{i=1}^{n} \sum_{j=1}^{n} \alpha_{i,j} y_{i,j} K\left(x_{i,j}\right), \tag{12}$$

where $0 \le \alpha_{i,j} \le C$, $\sum_{i=1}^{n} \alpha_{i,j} y_{i,j} = 0$.

Equation (12) represents the sum and quotient of all the results of the above equations in the case of the maximally combined solutions, obtaining *k* of the placed controllers.

Based on the SA searching process described above, multiple controllers are optimally placed in the SDN environment. In the above algorithm, the temperature points denote the optimal distance and location. The process is repeated until the optimal controller position is obtained. For the benefit of the reader, the pseudocode and a flow chart of the proposed SA-MCSDN algorithm are provided in Algorithm 2 and Figure 3, and Table 4 describes the abbreviations and parameters used therein.

**Table 4.** Parameters and abbreviations of SA-MSDN algorithm optimization.

| Parameter Name | Parameter Symbol | Value |
|---|---|---|
| Initial temperature | $T0$ | 10 |
| Final temperature | M | 0.00001 |
| No. of iterations at each temperature | $I_{max}$ | 5000 |
| Iteration | itr | 100 |
| Temperature decrementing factor (cooling rate) | $Alpha$ ($\alpha$) | 0.8 to 0.99 |
| solution | S | ? |
| Golden ratio (set empty) | $\varphi$ | ? |
| Temperature | T | ? |
| Probability | $Pr$ | ? |
| Temperature decline factor (random) | $\Theta$ | [0, 1] |
| Original reception rate | $\beta$ | 0.95 |
| Polynomial kernel | d | 2 and 3 |
| No. of nodes in the network | N | 3 topology (7, 28, 45) |
| [N × N] matrix consisting of shortest paths between every pair of nodes | D | $3 \times 3$ |
| Boltzmann constant | L | $1.380694 \times 10^{-23}$ J/K |
| No. of controllers to be deployed | $k$ | 21 |
| Array of size $k$ consisting of capacities of controllers | U | $3 \times 3$ |
| Array of size N consisting of demands of switches | V | $3 \times 3$ |
| Array of size $k$ consisting of the current positions of controllers | $Pc$ | ? |

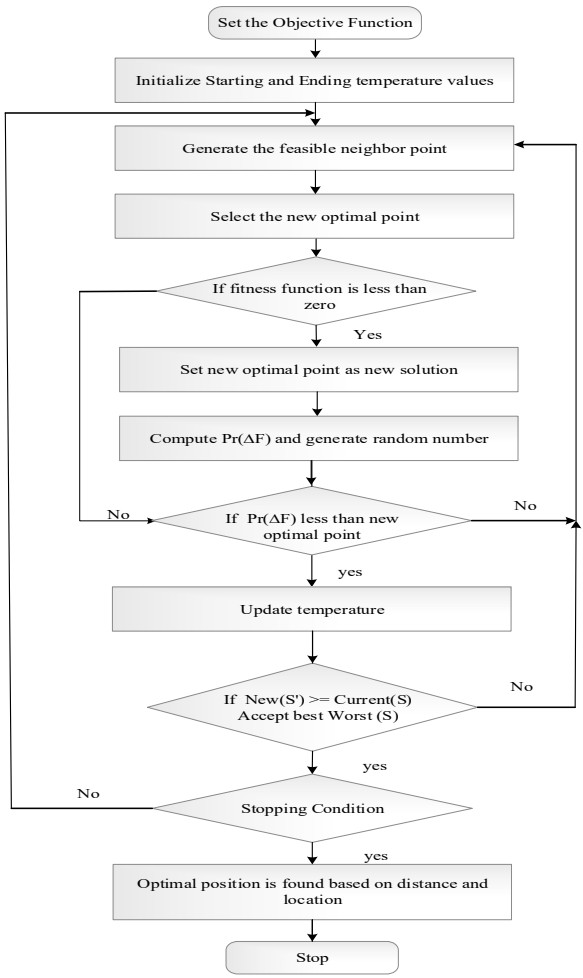

**Figure 3.** Flow chart of the SA-MCSDN algorithm.

---

**Algorithm 2** Pseudocode of the proposed SA-MCSDN algorithm

---

Initialized the objective function, F ($x_{i,j}$).
Initialize parameters, L, T, T0, and $\alpha$, $\beta$;
Set T = T0, V = 3 × D
Generate initial, S, $x_{i,j}$;
Calculate objective function, F ($x_{i,j}$);
S = F ($x_{i,j}$);
itr = 0;
  **While** (itr < $I_{max}$)

---

    optimal placement is not found **do**
      **For** a new optimal point
       Generate a feasible neighbor point New (S') of ($x_{i,j}$);
        **If** $\Delta F = (F(x'_{i,j})_{New} - F(x_{i,j})_{Current}) < 0$ **then**
        Set $F(x'_{i,j})_{New} = F(x_{i,j})_{Current}$;
        S ←S',
         S'← Neighbor(S)
         S'= F ($x'_{i,j}$)
         T← T0
          **Else**

          Compute $Pr(\Delta F) = exp^{\left(\frac{\Delta F}{T_{L(it\ )}}\right)}$

           Generate Θ;
         **If** (Θϵ[0,1]< Pr(ΔF) ) **then**
           S ←S', $\Delta F = (F(x'_{i,j})_{New} - F(x_{i,j})_{Current})$ ;
            Set = ($x'_{i,j}$) is an optimal position *(Pc)* of ($x_{i,j}$) ;
          **Else**
           $F(x'_{i,j})_{New} - F(x_{i,j})_{Current}$;
         **End if**
       **End if**
        After $I_{max}$ $T_L = ΘT_{L-1}$;
       **If** S' >= Worst Solution (S) **then**
        Accept best Worst Solution (S);
         **Else**

        Calculate T0 = $\frac{Worst - Best}{log(\beta)}$;

        $T_{itr+1} = \alpha × T_{itr}$; itr = itr + 1;
       **End if**
      **End For**
    **End While**

---

## 5. Experimental Results

In this section, we present the experimental results of the proposed SA-MCSDN algorithm, as well as a comparison with a previously proposed algorithm.

### 5.1. Simulation Setup

As shown in Figure 2, computers or devices are connected according to the proposed model architecture (MCSDN approach); the devices communicate and are connected

according to the implementation plan and according to the connection priority, respectively. The implementation plan includes the following steps:

**Step 1**: An initial NS3 is created with seven (*n*) controllers, twenty-eight (*n*) switches, and forty-five (*n*) users [32].

**Step 2**: Next, the topology is constructed based on an undirected graph, and the optimal controller selection process is performed using FA [33].

**Step 3**: Next, the MCPP process is performed using the proposed SA- MCSDN optimized algorithm.

**Step 4**: Sample packets are transitioned between the users.

**Step 5**: Finally, performance metrics, propagation delay, average round-trip time (RTT), matrix of time session (TS), average delay, reliability, throughput, cost, and fitness value are evaluated.

Table 5 outlines the framework arrangements, and Table 6 lists the setup parameters of the proposed MCSDN strategy.

**Table 5.** Original system configuration and input parameters using the SA-MCSDN algorithm.

| Software Specifications | Operating System | Windows 10 Pro (64 Bits) |
|---|---|---|
| | Type of topology | Undirected graph structure |
| | No. of Topologies | 3 |
| Hardware Specifications | Hard Disk | 1 T |
| | CPU | Intel(R) Core(TM) i7-10510U CPU @ 1.80 GHz 2.30 GHz |
| | RAM | 8.00 GB |

**Table 6.** Parameter configurations of a virtual machine and an SDN controller.

| VM Software and Hardware | Operating System | Linux Ubuntu-16.04 LTS -Desktop-Amd64 |
|---|---|---|
| | Integrated development environment (IDE) | VMware Workstation 16 Player |
| | Network simulator | NS3 |
| | Language | C++ |
| | Bandwidth | 100,000 downlink and uplink |
| | Hard disk | 20 GB |
| | RAM | 2048 MB |
| | Delay | 900 ms |
| | MIPS | 44,800 |
| Devices | Delay | 1 ms |
| | MIPS | 1500 |
| | RAM | 4 GB |
| SDN-CONTROLLER | No. of controllers | 7 |
| | No. of switches | 28 |
| | Switch delay | 5 µs |
| | Bandwidth | Variable |
| | Controller delay | 0.5 µs |
| | No. of hosts or users | 45 |
| | Packet size | 500 |
| | Static Const Unit SIM_DURATION | 200 |
| | speed | 299,792,458 |

When entering the information to run the simulation, the NS3 working steps are followed. Processing involves sending and receiving packets of data between switching devices and controllers, as well as among controllers themselves, as shown in Figure 4.

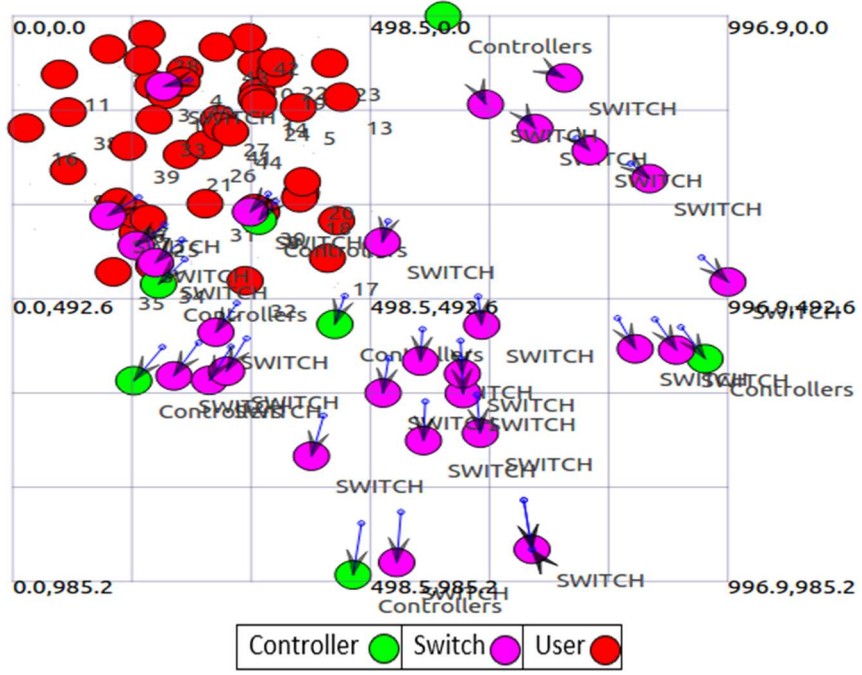

**Figure 4.** Multi-controller placement optimization by the SA-MCSDN algorithm.

After executing the NS3 network emulator, a software interface appears, consisting of an animator, stats, and packets. Figure 5 shows the process of exporting a table to optimally associate the chosen node location with other nodes. The table shown in Figure 5 consists of a group of device nodes, with the number of connections between these nodes ranging from 0 to 79, comprising 7 controllers, 28 switches, and 45 hosts. The table also contains ID and Mac addresses, through which devices are linked, as well as the optimal location for each controller after maximum iteration, which is correlated with the actual controller placement according to the improved SA-MCSDN algorithm.

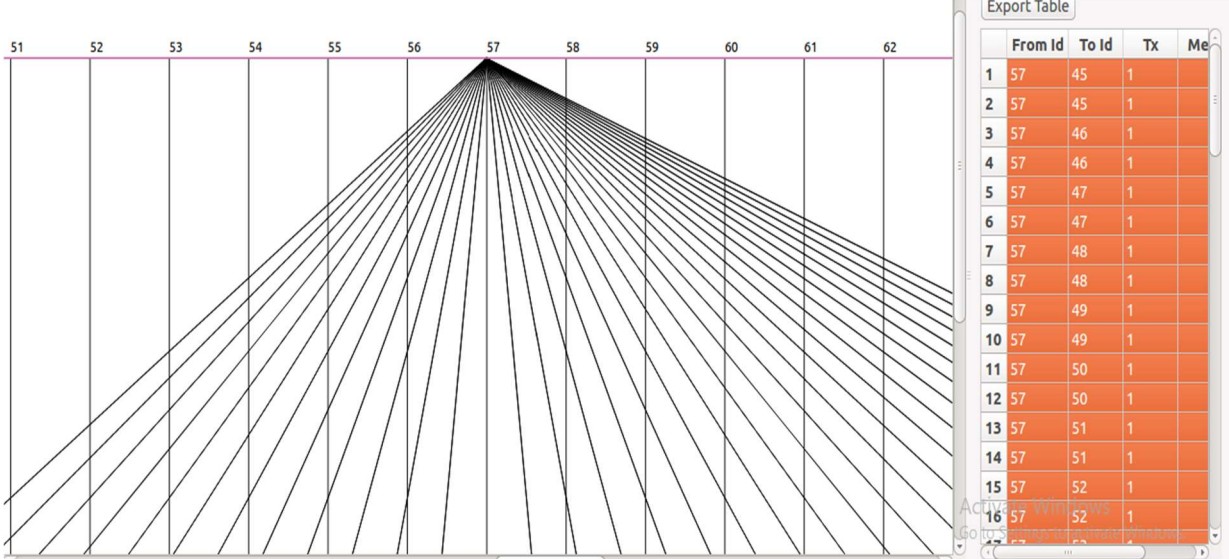

**Figure 5.** Optimal node placement relative to other nodes.

These graphical results represent a comparison of the SA-MCSDN algorithm proposed in this work with the HSA-PSO MCSDN algorithm proposed in [24] and SA-FFCCPP. The algorithm results were compared based on metrics, such as propagation delay, average round-trip time (RTT), matrix of TS, average delay, reliability, throughput, cost, and fitness value, as shown in the figures below. These figures show comparisons of the proposed SA-MCSDN algorithm and the previous HSA-PSO approach with respect to a number of SA-FFCCPP validation metrics. These graphical representations show that the proposed SA-MCSDN-based multi-controller placement approach in an SDN achieved improved results with respect to multiple metrics relative to the previous work using the hybrid HSA-PSO algorithm and SA-FFCCPP. The reason for the superior results is that SA-MCSDN enables a short execution time with considerable convergence, whereas the HSA-PSO algorithm requires a longer execution time with lower convergence and local optima traps, in addition to lacking robust management of exploitation and exploration. Our comparison of the SA-MCSDN algorithm with the previous hybrid HS-PSO algorithm revealed that the proposed SA-MCSDN performed better than the hybrid HS-PSO algorithm in terms of latency, propagation time, delay rate, reliability, defects, convergence, and cost. Table 2 presents a comparison of the average numerical results of the proposed method with those obtained in the previous work.

The following is a summary of the effects of the metrics through analysis and comparison with reference to the following figures and extracted results of the proposed algorithm.

Figure 6 shows the tradeoff between the algorithms, demonstrating the superiority of the proposed SA-MCSDN algorithm in terms of the impact of propagation delay.

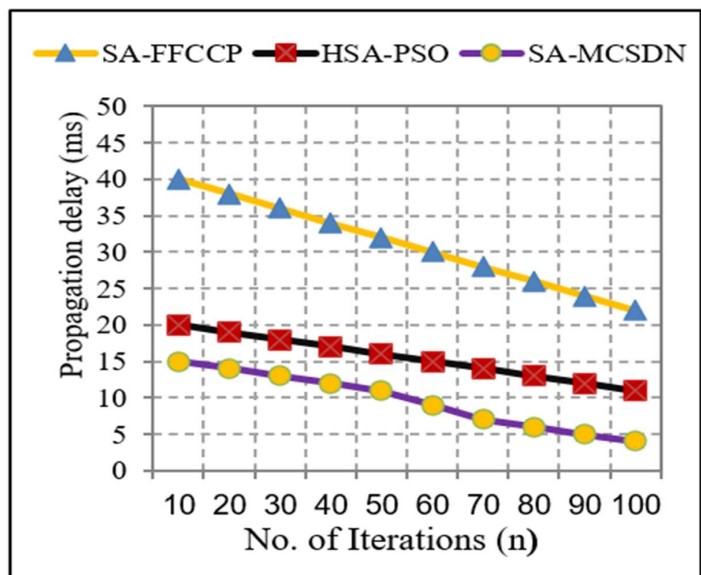

**Figure 6.** Comparison of propagation delay.

Latency is an important metric that is used to evaluate the delay between switches and controllers during propagation. It is calculated according to Equation (13):

$$\mathcal{P}_d = \frac{\lambda}{q} \tag{13}$$

where propagation delay ($\mathcal{P}_d$) is defined as the ratio between the distance ($\lambda$) (m) and the propagation speed ($q$) (m/s).

Figure 6 shows a comparison of the ($\mathcal{P}_d$) of the proposed SA-MCSDN method with the hybrid HSA-PSO and SA-FFCCP methods in terms of the number of iterations. An SDN network with low propagation delay attains efficient communication between S2Cs. The propagation delay decreases with an increasing number of iterations. Controller placement

(CP) was performed by considering only the distance between S2Cs. In addition, poor tuning of algorithms leads to poor CP, which increases the ($P_d$). In the proposed SA-MCSDN method, CP is performed by considering the ($\lambda$) using an optimization algorithm that increases the performance and reduces the ($P_d$). Figure 6 shows that the proposed SA-MCSDN method achieves a low propagation latency (4 to 15 ms) of 4.80 ms, compared with the hybrid HSA-PSO (11 to 20 ms; 15.6) and SA-FFCCPP (22 to 40 ms; 31.3) methods.

Figure 7 illustrates the tradeoffs between the algorithms listed above. The proposed SA-MCSDN algorithm achieves better performance than the HS-PSO and SA-FFCCP algorithms in terms of the impact of round-trip time (RTT; 6.05 ms).

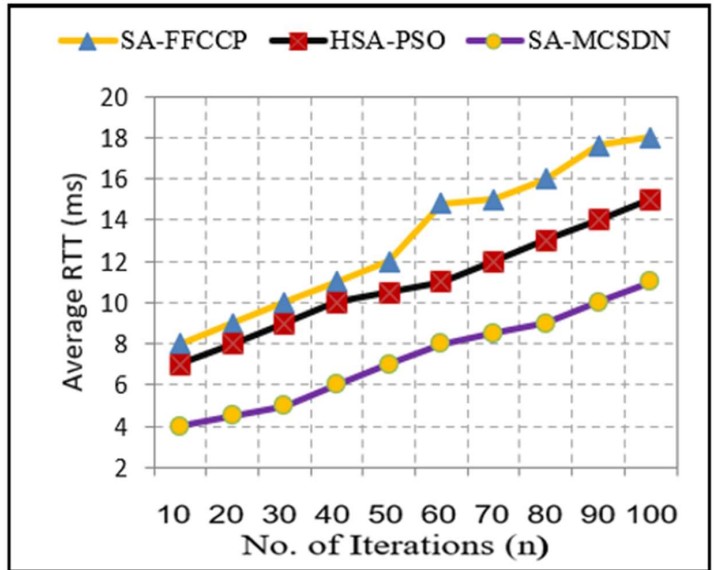

**Figure 7.** Comparison of average RTT.

Figure 8 shows a comparison of the algorithms listed above. The proposed SA-MCSDN algorithm achieves better performance than the HS-PSO and SA-FFCCP algorithms in terms of the effect of session time matrix (TS; 113.13).

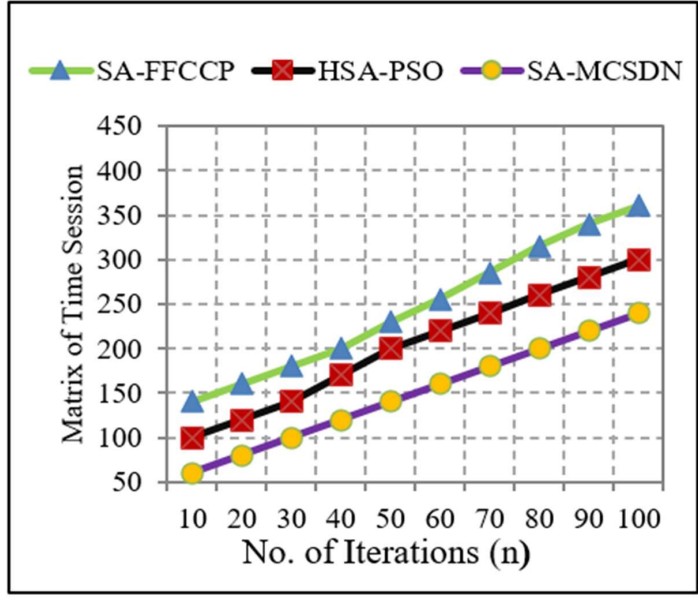

**Figure 8.** Comparison of the matrix of time session (TS).

Figure 9 illustrates a comparison of the algorithms listed above. The proposed SA-MCSDN algorithm achieves better performance than the HS-PSO and SA-FFCCP algorithms in terms of the average delay (42.57 ms).

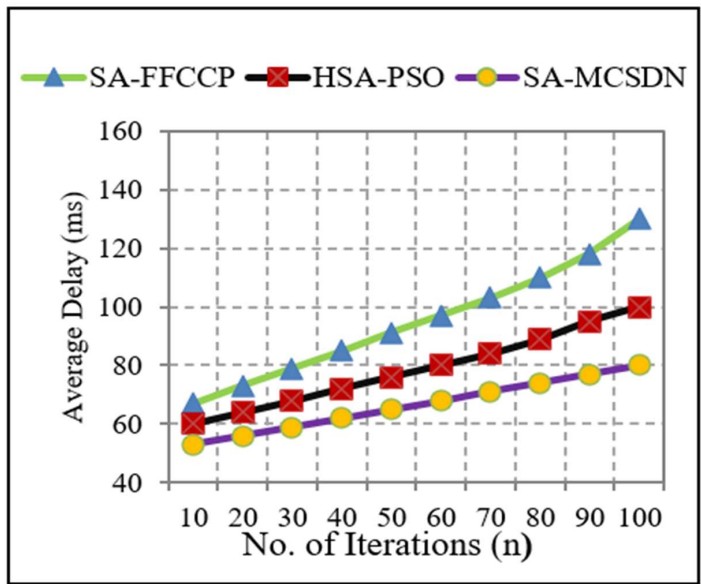

**Figure 9.** Comparison of average delay.

Figure 10 illustrates a comparison of the algorithms listed above. The proposed SA-MCSDN algorithm achieves better performance than the HS-PSO and SA-FFCCP algorithms in terms of the impact of reliability, which affects performance and flexibility, with a 99% ratio.

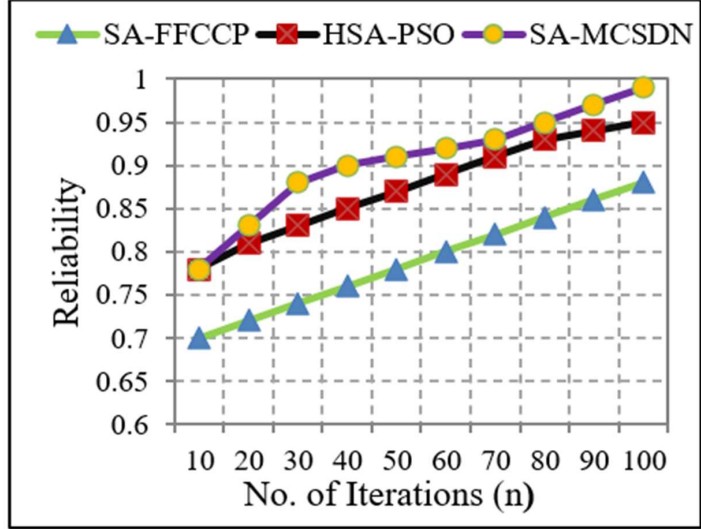

**Figure 10.** Comparison of reliability.

Figure 11 shows the tradeoff between the algorithms mentioned above. The proposed SA-MCSDN algorithm achieves better performance than the HS-PSO and SA-FFCCP algorithms in terms of the impact of throughput (395.57 Kbps).

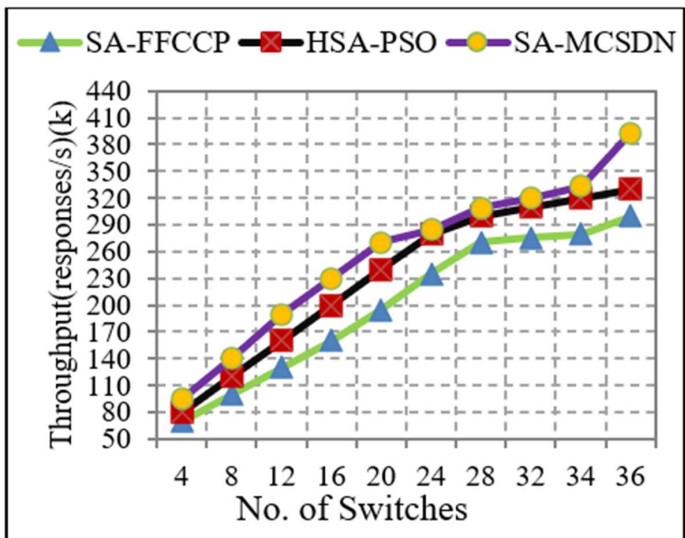

**Figure 11.** Comparison of throughput.

Figure 12 illustrates a comparison of the algorithms listed above. The proposed SA-MCSDN algorithm achieves slightly better performance than the SA-FFCCP algorithm and slightly worse performance than the hybrid HS-PSO algorithm in terms of the impact of the cost metric (31.27).

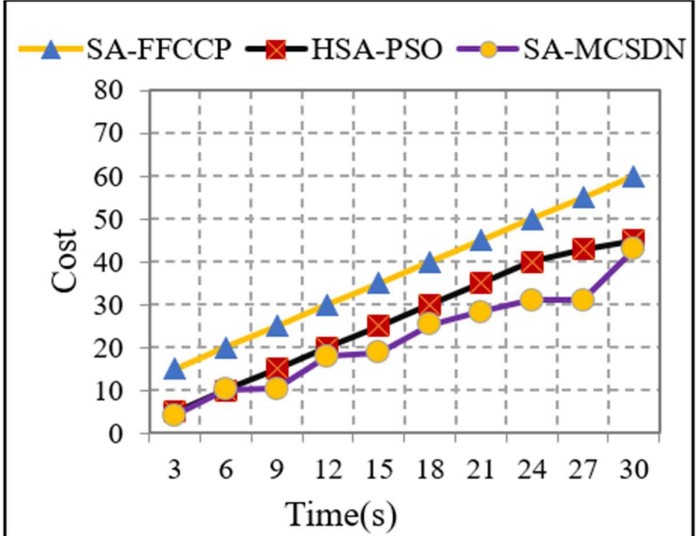

**Figure 12.** Comparison of cost.

Figure 13 denotes the convergence of the proposed algorithm with the previously proposed algorithms (to a large extent) in terms of the fitness value (Table 2).

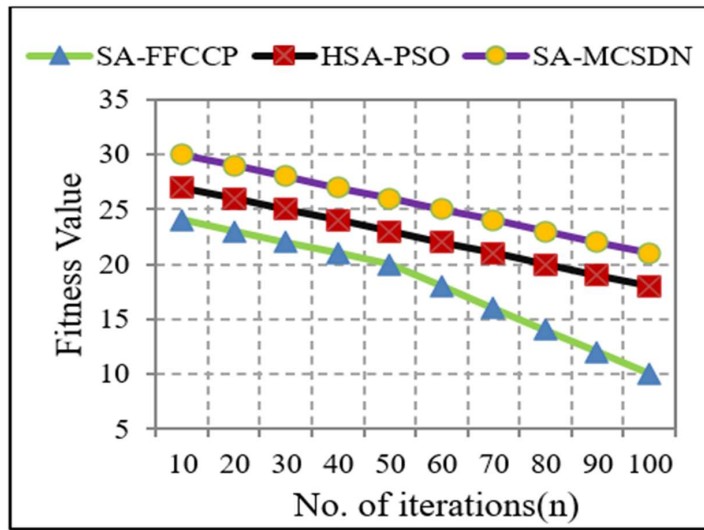

**Figure 13.** Comparison of fitness values.

*5.2. Comparative Analysis between HAS-PSO and SA-MCSDN Algorithms*

As shown in the table presented, our current work can be compared with our previous work; the results presented above confirm the existence of a large discrepancy in terms of performance improvement. According to the analysis of the schematics represented by the proposed SA-MCSDN, it achieves superior network performance relative to the previous hybrid HSA-PSO algorithm and the SA-FFCCPP algorithm in terms of scalability, reliability, delay in spread, and response. Furthermore, the quantitative amounts of the average qualitative metrics are mentioned according to the results extracted from the previous Figures for the proposed methods and respectively (propagation delay is 4.80 ms; round-trip time (RTT) is 6.05 ms; the matrix of time session (TS) is 113.13; the average delay is 42.57 ms; reliability is 99%; throughput is 395.57 Kbps; the cost is 31.27; and, finally, the fitness value is 19.16). Simulation of the multi-console mode is performed by the NS3 network simulator, and the simulation results show that the proposed work outperforms it. Table 2 shows the performance measures of the three algorithms in terms of scalability and other metrics.

**6. Conclusions and Future Work**

Through simulation experiments, we compared the SA-MCSDN algorithm with the SA-FFCCPP, HS-PSO, and SA-MCSDN algorithms using the NS3 simulator and an Ubuntu 64-bit tool in the SDN environment. We judged the improvement compared to our previously proposed algorithm. In order to increase the convergence rate and reduce deployment latency and connection latency, the following metrics were considered: propagation delay, round-trip time (RTT), matrix of time session (TS), average delay, reliability, throughput, cost, and fitness value. The proposed SA-MCSDN approach provides an improved solution to extend control over the SDN environment to Linux virtual machines, in addition to providing sufficient flexibility to handle such environments. Additional studies on this topic are necessary.

We found that the proposed SA-MCSDN algorithm achieves better results than our previously proposed algorithm in solving the problem of distributing control units and placing them in optimal and appropriate positions in terms of spatial location within the network in the control layer, as well as the time taken for implementation. This analysis is based on the following main criteria: the shortest distance and the method of its calculation, the decrease in the time taken for distribution, and the communication between devices of the infrastructure layer and the control units (S2Cs), as well as between the control units themselves (C2Cs) with consideration of fault. The selection and preparation of control units and the measures of performance shown in Table 2 indicate the level of effectiveness

of the network, the increase in the expansion of the network, and its reliability. In future studies, we intend to emphasize and identify various drawbacks of a variety of approaches.

**Author Contributions:** Conceptualization, N.S.R., S.T.F.A.-J. and K.S.J.; methodology, N.S.R., S.T.F.A.-J. and K.S.J.; software, N.S.R., S.T.F.A.-J. and K.S.J.; validation, N.S.R., S.T.F.A.-J. and K.S.J.; formal analysis, N.S.R., S.T.F.A.-J. and K.S.J.; investigation, N.S.R., S.T.F.A.-J. and K.S.J.; resources, N.S.R., S.T.F.A.-J. and K.S.J.; data curation, N.S.R., S.T.F.A.-J. and K.S.J.; writing—original draft preparation, N.S.R., S.T.F.A.-J. and K.S.J.; writing—review and editing, N.S.R., S.T.F.A.-J. and K.S.J.; visualization, N.S.R., S.T.F.A.-J. and K.S.J.; supervision, N.S.R., S.T.F.A.-J. and K.S.J.; project administration, N.S.R., S.T.F.A.-J. and K.S.J.; funding acquisition, N.S.R., S.T.F.A.-J. and K.S.J. All authors have read and agreed to the published version of the manuscript.

**Funding:** This research received no external funding.

**Institutional Review Board Statement:** Not applicable.

**Informed Consent Statement:** Not applicable.

**Data Availability Statement:** All data are presented in the main text.

**Acknowledgments:** We would like to thank all individuals and organizations who provided support to complete this paper.

**Conflicts of Interest:** The authors declare no conflict of interest.

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
