# Peer review of "Using Metaheuristics (SA-MCSDN) Optimized for Multi-Controller Placement in Software-Defined Networking"

_futureinternet, doi:10.3390/fi15010039_

Round 1
Reviewer 1 Report
Authors propose a guiding model called Simulated Annealing for Multi Controllers in SDN algorithm to solve the problem of placing multiple controllers in the appropriate location with estimation of distance considerations and times of distribution between controllersas well as between controllers and switches.
There are some issues that must be fixed before accepting the paper:
- Figure 1 seems to belong to other paper. It must be removed. Moreover, it has no sense to place a figure after the explanation of the rest of the paper in the introduction section.
The related work section should have also an explanation of what should be inside.
I have seen some related work not cited that should be cited (Moreover there are few references for a journal paper):
Software Defined Network-based control system for an efficient traffic management for emergency situations in smart cities, Future Generation Computer Systems 88, 243-253. 2018
A QoS-Based routing algorithm over software defined networks, Journal of Network and Computer Applications 194, 103215. 2021
Algorithm 1 must be cited from the text.
Figure 4 is not cited from the text.
Figure 7 to Figure 10 have 2 graphs, each graph must be 1 figure. Each figure must be explained in detail.
Author Response
Response Report
Response to Referee No. 1 Comments:
Dear Prof.
First, we would like to express our thanks and gratitude for your great and valuable comments. Your comments help us to improve the quality and usefulness of our paper. Regarding your comments, we have been doing the following changes:
- Figure 1 seems to belong to other paper. It must be removed. Moreover, it has no sense to place a figure after the explanation of the rest of the paper in the introduction section.
Response: We have deleted Figure 1 and manage the numbering of the figures.
The related work section should have also an explanation of what should be inside.
Response: We have added explanation from Reference [24] and Table 1 in section ( 2. Related Work).
I have seen some related work not cited that should be cited (Moreover there are few references for a journal paper):
Software Defined Network-based control system for an efficient traffic management for emergency situations in smart cities, Future Generation Computer Systems 88, 243-253. 2018
A QoS-Based routing algorithm over software defined networks, Journal of Network and Computer Applications 194, 103215. 2021
Response: We have added) References [22] and [23]) in ( 2. Related Work)
Algorithm 1 must be cited from the text.
Response: We have added Algorithm 1 citation from the text.
Figure 4 is not cited from the text.
Response: We have added Figure 4 citation from the text.
Figure 7 to Figure 10 have 2 graphs, each graph must be 1 figure. Each figure must be explained in detail.
Response: We have revised the figure (from Figure 6 to Figure 13 in the revised manuscript) each alone with one graph and we explained them

Reviewer 2 Report
In this article, the authors propose a SA-MCSDN guided model to solve the issue of placing multiple controllers in appropriate locations. Some issues identified in earlier studies were resolved in the study. It's a good idea to try to address those dilemmas through models.
After rereading it, I am confused by these points. Perhaps the author can reinforce the explanation within the article, so that readers of the article can more clearly resolve the reading difficulties.
1. How does Figure 3 relate to the equations outlined in Chapter 4? Although briefly mentioned on line 132 of the text, the explanation needs to be clearer.
2. In chapter 4.3, I think the description of the proposed algorithm is somewhat fuzzy. Can you elaborate on why these formulas are used step by step and why these formulas derive from properties not just step by step. A more specific explanation is to solve these problems, we have devised such an equation, which can mainly solve these problems It is done by maxminze Extract something.
3. 5.1 Can you refer to image 3 and explain the number of computers you used for simulation? Is this architecture comparable with Figure 3?
4. The controller in Figure 5 seems to be green? Is it wrong?
5. It is suggested that Figure 6 be clearly explained.
In general, the authors conducted experiments to verify the arguments put forward by the simulations. It's an excellent piece of work. Experience shows that the verification of the conclusion is clear and the description of the process is clear. For the benefit of the reader, this article must be clarified.
Author Response
Response to Referee No. 2 Comments:
Dear Prof.
First, we would like to express our thanks and gratitude for your great and valuable comments. Your comments help us to improve the quality and usefulness of our paper.
Regarding your comments, we have been doing the following changes:
- How does Figure 3 relate to the equations outlined in Chapter 4? Although briefly mentioned on line 132 of the text, the explanation needs to be clearer.
Response: We have added an explanation of Figure 3 for the purpose of linking it to the equations in Chapter 4.
- In chapter 4.3, I think the description of the proposed algorithm is somewhat fuzzy. Can you elaborate on why these formulas are used step by step and why these formulas derive from properties not just step by step. A more specific explanation is to solve these problems, we have devised such an equation, which can mainly solve these problems It is done by maxminze Extract something.
Response: We have added an explanation and clarification of all equations in chapter (4.3 Multi-Controller Placement Using Simulated Annealing Algorithm)
- 5.1 Can you refer to image 3 and explain the number of computers you used for simulation? Is this architecture comparable with Figure 3?
Response: We have added and referred to the figure 3 with the addition of the proposed implementation plan for connecting the computers, and we have detailed the preparations for connecting them.
- The controller in Figure 5 seems to be green? Is it wrong?
Response: We have used green for Controller in Figure 5.
- It is suggested that Figure 6 be clearly explained.
Response: We have added an explanation for Figure 6.
In general, the authors conducted experiments to verify the arguments put forward by the simulations. It's an excellent piece of work. Experience shows that the verification of the conclusion is clear and the description of the process is clear. For the benefit of the reader, this article must be clarified.
Response: We have added clarifications throughout the paper.

Reviewer 3 Report
A relevant paper entitled "Using metaheuristics (Simulated Annealing for Multi Controllers in software-defined networks) optimized for multi-controllers' placement in software-defined networking."
Please, correct the typos and/or grammatical errors.
The abstract should be extensively revised to reflect what was investigated (the problem); how it was investigated (the methodology); and the performance metrics (results and implications).
The relevant qualitative and quantitative metrics should be judiciously stated in the abstract and the conclusion.
The relevant existing works have been studied to delineate the reported research work but the novelty of the research findings is marginal.
The authors should explain what they mean by “optimal and appropriate position” respect to the spatial and temporal resolutions of the propsoed technique.
Please, provide a detailed analysis of the latency of the distribution and the communication between the devices of the infrastructure layer and the control units. What are the design and performance considerations for latency and distance optimisation for similar devices within the software-defined networks?
Author Response
Response to Referee No. 3 Comments:
Dear Prof.
First, we would like to express our thanks and gratitude for your great and valuable comments. Your comments help us to improve the quality and usefulness of our paper.
Regarding your comments, we have been doing the following changes:
Comments and Suggestions for Authors
Please, correct the typos and/or grammatical errors.
Response: We have corrected the typos and grammatical errors. And sent article to address link : https://www.mdpi.com/authors/english for editing .
The abstract should be extensively revised to reflect what was investigated (the problem); how it was investigated (the methodology); and the performance metrics (results and implications).
Response: We have revised the Abstract to clarify the solution of the problem, the used methodology, and verifying it through performance metrics according to the extracted results.
The relevant qualitative and quantitative metrics should be judiciously stated in the abstract and the conclusion.
Response: We have added qualitative and quantitative metrics in the abstract and the section (6. Conclusion and Future Work).
The relevant existing works have been studied to delineate the reported research work but the novelty of the research findings is marginal.
Response: We have added more clarification to better reflect the novelty of the research findings.
The authors should explain what they mean by “optimal and appropriate position” respect to the spatial and temporal resolutions of the propsoed technique.
Response: We have clarified the meaning of an "optimal and appropriate position" in Section (6. Conclusions and Future Work)
Please, provide a detailed analysis of the latency of the distribution and the communication between the devices of the infrastructure layer and the control units. What are the design and performance considerations for latency and distance optimisation for similar devices within the software-defined networks?
Response: We have added detailed analysis of the latency in section( (5. Experimental Results ) (5.1. Simulation Setup)) for (Figure 6. Comparison of propagation delay)

Reviewer 4 Report
Dear Authors
Your proposal is very interesting, SDN currently has a very important role in virtualization and cloud computing management.
My suggestions:
1. improve the wording of the whole document.
2. Change the way of citing research, read other papers and look at the style and way of citing, it is not as common as you do (Reference [2]), for example, you could write the authors of [2], or reference using the surname of the authors.
3. Check the title of the tables, the previous text is very repetitive, and this isn't very clear. For example your paper on line 110 onwards:
Below is Table 1. That shows the weaknesses of the previous work that used the SA algorithm, which was processed by the proposed algorithm (SA-MCSDN) in terms of comparing performance measures:
- Table 1. Tables should be placed in the main text near to the first time they are cited.
4. Explain better the methodology used in this research, and clarify if this current research is the continuation of previous works, which you have already published because there are some similarities.
5. You say that your work is inspired by "The SA is inspired from the thermodynamic process, during reducing the energy of material, it changes at state to lower levels". Then you should explain in a minimum section, about this mechanism.
6. In section 5 "Experimental Results", step 1 refers to parameters from other investigations 30 and 31". It would be better if you write these parameters directly in this section, they are very confusing.
7. Explain the results in a better way, e.g. better describe the results of Figure 6 and the others.
8. Improve the wording of the conclusions, they refer to other research.
Author Response
Response to Referee No. 4 Comments:
Dear Prof.
First, we would like to express our thanks and gratitude for your great and valuable comments. Your comments help us to improve the quality and usefulness of our paper.
Regarding your comments, we have been doing the following changes:
improve the wording of the whole document.
Response: The wording of the whole document has been improved. And sent article to address link : https://www.mdpi.com/authors/english for editing .
- Change the way of citing research, read other papers and look at the style and way of citing, it is not as common as you do (Reference [2]), for example, you could write the authors of [2], or reference using the surname of the authors.
Response: We have fixed this issue in section (2. Related work).
- Check the title of the tables, the previous text is very repetitive, and this isn't very clear. For example, your paper on line 110 onwards:
Below is Table 1. That shows the weaknesses of the previous work that used the SA algorithm, which was processed by the proposed algorithm (SA-MCSDN) in terms of comparing performance measures:
- Table 1. Tables should be placed in the main text near to the first time they are cited.
Response: We have fixed this issue and added appropriate title of Table 1.
- Explain better the methodology used in this research, and clarify if this current research is the continuation of previous works, which you have already published because there are some similarities.
Response: We have added a clear explanation of the research methodology in Section (4. Proposed Work) and have made it clearer that this current research is a continuation of our previous work.
- You say that your work is inspired by "The SA is inspired from the thermodynamic process, during reducing the energy of material, it changes at state to lower levels". Then you should explain in a minimum section, about this mechanism.
Response: We have added explanation of the same section (4.3. Multi-Controller Placement Using Simulated Annealing Algorithm)
- In section 5 "Experimental Results", step 1 refers to parameters from other investigations 30 and 31". It would be better if you write these parameters directly in this section, they are very confusing.
Response: We have added parameters in Section (5. Experimental Results) (5.1. Simulation Setup) step 1.
- Explain the results in a better way, e.g. better describe the results of Figure 6 and the others.
Response : We have added an explanation about Figure 6 and the figures after it.
- Improve the wording of the conclusions, they refer to other research.
Response: We have added an explanation on the topic and improved the wording of conclusions in the section ( 6. Conclusion and Future Work)
Thanks

Round 2
Reviewer 1 Report
Authors have fixed all my comments.
Author Response
Dear Prof.
First, we would like to express our thanks and gratitude for your great and valuable comments. Your comments help us to improve the quality and usefulness of our paper.
Thanks
Reviewer 2 Report
Comments given are addressed well
Author Response

(The authors gave the same response as above.)

Reviewer 3 Report
The authors have stated the qualitative performance metrics thus: "The, and these metrics considered in this work are propagation delay, round- trip time (RTT), the matrix of time session 33 (TS), average delay, reliability, throughput, cost, and fitness value"
Please, state the quantitative performance metrics (numbers for each of the mentioned parameter.
Author Response
Dear Prof.
First, we would like to express our thanks and gratitude for your great and valuable comments. Your comments help us to improve the quality and usefulness of our paper. Regarding your comments, we have been doing the following changes:
The authors have stated the qualitative performance metrics thus: "The, and these metrics considered in this work are propagation delay, round- trip time (RTT), the matrix of time session 33 (TS), average delay, reliability, throughput, cost, and fitness value"
Please, state the quantitative performance metrics (numbers for each of the mentioned parameter.
Response: We have added the quantitative performance metrics (numbers for each of the mentioned parameter) in section (5. Experimental Results )( 5.2. Comparative Analysis between HAS-PSO and SA-MCSDN algorithms)
Thanks
Reviewer 4 Report
Dear Authors
It is much better.
Author Response

(The authors gave the same response as above.)
